# Why Do Insects Close Their Spiracles? A Meta-Analytic Evaluation of the Adaptive Hypothesis of Discontinuous Gas Exchange in Insects

**DOI:** 10.3390/insects13020117

**Published:** 2022-01-22

**Authors:** Seun O. Oladipupo, Alan E. Wilson, Xing Ping Hu, Arthur G. Appel

**Affiliations:** 1Department of Entomology and Plant Pathology, Auburn University, Auburn, AL 36849, USA; huxingp@auburn.edu (X.P.H.); appelag@auburn.edu (A.G.A.); 2School of Fisheries, Aquaculture, and Aquatic Sciences, Auburn University, Auburn, AL 36849, USA; wilson@auburn.edu

**Keywords:** gas pattern, ${\dot{V}}_{CO_{2}}$, metabolic rate, DGE, DGC, hygric, chthonic, hexapod, respiratory physiology

## Abstract

**Simple Summary:**

Insects breathe with the aid of thin capillary tubes that open out to the exterior of their body as spiracles. These spiracles are often modulated in a rhythmic gas pattern known as the discontinuous gas exchange cycle. During this cycle, spiracles are either firmly shut to allow no gaseous exchange or slightly open/fully open to allow for gaseous exchange. Two explanations are put forward to rationalize this process, namely, the rhythmic pattern is to (1) reduce water loss or (2) facilitate gaseous exchange in environments with high carbon dioxide and low oxygen. Interestingly, certain insects (such as some desert insects) do not use this rhythmic pattern where it would have been most beneficial and logical. Such an observation has led to the questioning of the explanations of the discontinuous gas exchange cycle. Consequently, we attempt to resolve this controversy by conducting a meta-analysis by synthesizing apposite data from across all insects where a discontinuous gas exchange cycle has been reported. A meta-analysis allows for a shift from viewing data through the lens of a single species to an order view. Thus, our goal is to use this holistic view of data to examine the explanations of the discontinuous gas exchange cycle across multiple groups of insects.

**Abstract:**

The earliest description of the discontinuous gas exchange cycle (DGC) in lepidopterous insects supported the hypothesis that the DGC serves to reduce water loss (hygric hypothesis) and facilitate gaseous exchange in hyperoxia/hypoxia (chthonic hypothesis). With technological advances, other insect orders were investigated, and both hypotheses were questioned. Thus, we conducted a meta-analysis to evaluate the merit of both hypotheses. This included 46 insect species in 24 families across nine orders. We also quantified the percent change in metabolic rates per °C change of temperature during the DGC. The DGC reduced water loss (−3.27 ± 0.88; estimate ± 95% confidence limits [95% CI]; *p* < 0.0001) in insects. However, the DGC does not favor gaseous exchange in hyperoxia (0.21 ± 0.25 [estimate ± 95% CI]; *p* = 0.12) nor hypoxia, but did favor gaseous exchange in normoxia (0.27 ± 0.26 [estimate ± 95% CI]; *p* = 0.04). After accounting for variation associated with order, family, and species, a phylogenetic model reflected that metabolic rate exhibited a significant, non-zero increase of 8.13% (± 3.48 95% CI; *p* < 0.0001) per °C increase in temperature. These data represent the first meta-analytic attempt to resolve the controversies surrounding the merit of adaptive hypotheses in insects.

## 1. Introduction

Insects modulate (Oxygen) O_2_ and (carbon dioxide) CO_2_ by a means of internal air-filled tracheae. The tracheal system ramifies into progressively thinner branches to deliver O_2_ and remove CO_2_ as a waste product of cellular respiration from every cell in the insect body [1,2]. Externally, the tracheae open out as spiracles that are positioned laterally along the insect body. Typically, insects have zero or one pair of spiracles per segment, with a maximum number of 10 pairs on an adult insect [2,3]. Some basic variations abound in the number, role, and sophistication of spiracles in insects depending on the species and stage of development [3]. For example, adult cockroaches have simple tracheae with valve spiracles located laterally along the abdomen, mosquito larvae have one functional terminal spiracle, and most endoparasitic larvae have a closed tracheal system with cutaneous gas exchange [2]. Regardless of the organism and the respiratory medium, gaseous exchange is always through either convection (i.e., bulk flow) and/or diffusion (i.e., movement from a higher concentration to a lower concentration region) [4]. This is true because while atmospheric pressure at sea level is 760 mmHg (101.33 kPa), the atmospheric volume of O_2_ (~21%) and CO_2_ (~0.04%) creates a partial pressure gradient between the atmosphere and an organism’s interior. In other words, the partial pressure (*PO_2_*) of oxygen and carbon dioxide (*PCO_2_*) is 1590 mm Hg (21.28 kPa) and 0.30 mmHg (0.04 kPa), respectively. Based on these calculations and the partial pressure difference in insects, it is easy to see why O_2_ readily diffuses in and CO_2_ diffuses out of any living system. However, insect respiration is not that straightforward. This is because gaseous exchange can be influenced by the environment and metabolic demands. Consequently, insects may employ more than one type of gas exchange pattern. Interestingly, it is not uncommon to have the same insect employ more than one gas exchange pattern over a period [5]. This begs the question: what advantage could there be to the choice of one pattern and abandonment of another?

To date, most measures of gas exchange patterns in insects rely on the measurement of CO_2_ and not O_2_, because the latter is more difficult to measure, whereas CO_2_ can be measured much more accurately [6]. As such, unsurprisingly, the flow-through respirometry is optimized for CO_2_ readings. Hence, the gas patterns in insects are described based on the state (efflux/influx) of CO_2_. In any case, there is a consensus on three gas patterns in insects (Figure 1). A continuous gas exchange in which there is an irregular efflux of CO_2_, a discontinuous gas exchange (DGC) in which there is a periodic burst of CO_2_ separated by intervals of low/negligible CO_2_ release, and a cyclic gas exchange pattern in which there is a regular increase of CO_2_ but separated intervals with minimal CO_2_ emission are lacking [7,8,9,10,11]. Gaseous exchange in insects is established by inward diffusion (or in some cases convection) of O_2_ via the spiracles to the tracheae and cells, and the outward net movement of CO_2_ (and water vapor molecules) is often regarded as uncoupling of O_2_ consumption and CO_2_ emission. Thus, it is the characteristic of the spiracles at a given period that creates the three distinct gas exchange patterns: a discontinuous gas exchange cycle (DGC), cyclic gas exchange, and continuous gas exchange [12].

Of these gas exchange patterns, the DGC has received the most attention, possibly due to the presence of an F-phase and its implications [13]. Classical experiments involving the diapausing pupae of the *Cecropia* moth revealed the phases of a DGC [14]. It starts with a closed phase (i.e., C-phase), where O_2_ consumption by the tissues lowers its endotracheal O_2_ partial pressure (i.e., *PO_2_*) such that the organism’s total endotracheal pressure is lowered, and the extracellular fluid buffers CO_2_ (Figure 1c). When the O_2_ level in the organism substantially drops, the spiracles flutter, rapidly opening and closing, allowing the flow of O_2_ in the air into the organism. This is the flutter phase (i.e., the F-phase). This continues until the level of CO_2_ becomes high in the tracheal system and triggers the spiracular valves to open widely to allow for CO_2_ emission and additional O_2_ uptake [5,6]. This is the open phase (O-phase). The cyclic gas exchange consists of a regular burst of gas exchange and no prolonged C-phase, while an O-phase dominates a continuous gas exchange pattern producing irregular curves [15].

As highlighted earlier, the questions surrounding the significance of the DGC are centered on the understanding of the F-phase. For example, the hygric hypothesis argues that the DGC serves to reduce respiratory water loss [7,14]. This hypothesis is strengthened by the water and CO_2_ retention during the F-phase. It assumes that during the F-phase, in which the spiracles flutter to allow for gaseous exchange, the flow of air is convectional, and thus, only O_2_ uptake occurs. If otherwise (i.e., bidirectional), the simultaneous intake of O_2_ and emission of CO_2_ would exhibit concomitant water loss. This is true because tracheole gases are saturated with water vapor; thus, emitted CO_2_ would have incurred a water-loss penalty. Another hypothesis argues that the DGC serves to facilitate gaseous exchange in hypercapnic and hypoxic conditions [16]. Again, this hypothesis relies on the F-phase. In this case, it is believed that some insects might find themselves or might predominantly occupy a habitat with high CO_2_ (hypercapnic)—as is the case with subterranean termites and ants with nests underground—or low O_2_ (hypoxic). Thus, by buffering CO_2_, for an extended period, the internal *PCO_2_* can be high enough so that external *PCO_2_* and CO_2_ emission can follow the concentration gradient by diffusion away from the insect. Similarly, by consumption of O_2_ internally, the O_2_ becomes lower than external *PO_2_* and O_2_ can easily flow in by diffusion. Interestingly, for this hypothesis to be also true, the F-phase has to be convectional; otherwise, the hypothesis becomes weakened.

Importantly, the three phases of the DGC provide a mechanistic way of comparing and evaluating hypotheses. For example, by comparing cuticular (interburst; CF-phases) to respiratory (burst; O-phase) water loss, the DGC is argued to minimize respiratory water loss [7,17,18]. By modulating spiracular closure sessions, the DGC is thought to enhance gaseous exchange in hyperoxic/hypoxic environments [16] or prevent O_2_ toxicity [19]. Additionally, it is often argued that the DGC is expressed to regulate metabolic demand [13,20], but there is no quantification of how temperature influences metabolic rate during the DGC beyond two temperatures. However, we know that metabolic and temperature rates can be driving forces of the DGC [21].

It is noteworthy to add that other adaptive (e.g., the strolling arthropod hypothesis and oxidative damage hypothesis) and non-adaptive (e.g., the emergent properties hypothesis and neural hypothesis) hypotheses have been posited to explain the occurrence of the DGC (see review by [13,15]). Nevertheless, the focus of this meta-analytic study is on hypotheses that measure CO_2_, O_2_, and water from rhythmic spiracular closure and opening during the DGC. To date, there is a controversy surrounding the acceptance of these hypotheses from one insect clade to another (see the extensive and excellent reviews by [15,22,23]). Such contention may be because quantitative comparisons across clades (i.e., diverse taxa) to allow for a conclusion are lacking. To compare, it might be helpful to utilize an approach robust enough to allow for the synthesis of results across taxa (and/or studies) while maintaining not just the statistical significance but the biological relevance of individual studies. Statistically speaking, such an approach would highlight the magnitude of the finding(s) from each study (regardless of the inference drawn from such data) and resolve to find an overarching theme across studies. Indeed, such a comparison would require the synthesis of results across life stages, species (and possibly geographically isolated species), families, orders, habitats, and experimental conditions. In short, by synthesizing results from published data on the DGC across taxa, one might be able to resolve the conundrum surrounding the adaptive significance of the DGC on a broader scale.

Synthesizing results across studies is not a new approach; narrative reviews essentially do that. However, narrative reviews, at best, gather evidence together and fail to synthesize the evidence transparently and objectively. Thus, it follows that most narrative reviews are not reproducible and bias-laden. To circumvent this, a meta-analytic approach evaluates the estimates of magnitude or effects of interest (i.e., effect sizes) from each study beyond significance testing (as done with a *p*-value) using a quantitative approach [24]. In other words, a meta-analysis goes beyond the dichotomy of a *p*-value (i.e., reject or fail to reject the null) to determine the magnitude and precision of the estimate (i.e., effect size) from each study and converts this estimate to a standardized metric [25,26]. This conversion is critical given the variability in study designs and outcomes towards a given question, and thus, creates a metric to facilitate comparison of outcomes across studies. This makes a meta-analytical approach transparent, reproducible, and updatable. The reproducibility of a meta-analytical study is well established by the reporting guidelines of PRISMA (Preferred Reporting Items for Systematic Reviews and Meta-Analyses; http://www.prisma-statement.org/ last accessed 15 December 2021). PRISMA is “an evidence-based minimum set of items for reporting in systematic reviews and meta-analyses” and includes a checklist and a template flowchart to reflect the path an investigator employs to arrive at study selection. In short, a meta-analysis provides a more powerful and less biased means for clarifying, quantifying, synthesizing, and disproving (or confirming) assumed wisdom than do conventional approaches. Moreover, it is an index of the biological importance of individual study as opposed to statistical importance from null-hypothesis testing [25]. Importantly, a meta-analysis is a powerful tool for evidence appraisal especially when controversies exist.

The meta-analytic design of this study is an attempt to create a broad-scale comparison to evaluate the merit of some of the adaptive hypotheses described to account for the significance of the DGC in insects. We propose that for a given DGC, the metabolic rate (${\dot{V}}_{CO_{2}}$) is most likely an intrinsic component that gives the relevance of one DGC phase to another across different taxa. That is, this component may help answer the question of why insects close their spiracles under a given condition. Additionally, it is pertinent to ask further questions by taking a closer look at insects in general exhibiting a DGC. For example, does rhythmic spiracular closure maintained to reduce respiratory water loss in insects? Does a DGC facilitate gas exchange in hypoxia and hyperoxia? How much does the metabolic rate change with a degree change in temperature for insects breathing during a DGC?

Consequently, the objectives of this study were to (1) evaluate if the DGC serves to reduce respiratory water loss in insects, (2) determine if the DGC facilitates gaseous exchange under chthonic conditions, and (3) quantify the percent change in metabolic rates per °C increase in temperature during the DGC. This study is the first attempt to employ a quantitative meta-analytic approach to evaluate the merit of adaptive hypotheses associated with the DGC across taxa.

## 2. Materials and Methods

Study selection for this meta-analysis was done according to the Preferred Reporting Items for Systematic Reviews and Meta-Analyses statement (PRISMA; http://prisma-statement.org/PRISMAStatement/CitingAndUsingPRISMA last accessed 15 December 2021) (Figure 2).

### 2.1. Search Strategy

Literature searches were conducted in the English language on 10 January 2020. Searches were conducted using Web of Science and PubMed with the following terms: “gas patterns*”, “discontinuous gas exchange”, “DGC*”, “DGE” “cuticular water loss”, “respiratory water loss”, and “gas exchange patterns”. The title and abstract for each paper (in both search engines) were screened for relevancy. Duplicates and papers that were not DGC- or DGE-specific were removed. A study was selected if it satisfied the inclusion criteria for at least one objective.

### 2.2. Inclusion Criteria and Data Extraction

One criticism of meta-analyses is the number of stringent rules regarding paper selection; thus, we were less stringent with the exclusion criterion, as demonstrated by the variability in treatment types of the selected studies. Papers were selected based on the assessment of the gas pattern by the author as either a DGC or DGE. In order of perceived significance, the coefficient of variation CO_2_ emission over the entire DGC,$\;{\dot{V}}_{CO_{2}}$ across DGC phases, the volume of burst CO_2_ emissions, the duration of burst CO_2_ emissions, the duration of interburst CO_2_ interval, and the DGC frequency are the relevant indices (Lighton, personal communication). Given that we were able to find more datasets with$\;{\dot{V}}_{CO_{2}}$, we decided to use this index. Additionally, all selected papers reported the flow rate, experimental temperature—either at a specific temperature or over a range of temperatures (if corresponding metabolic rates were recorded)—mean mass (±SD or SE), and the number of insects investigated (n). To be included in objective 1 (does the DGC reduce respiratory water loss in insects?), a study needed to report mean cuticular and respiratory water loss and some form of measurement variance (SD or SE). Consequently, a comparison was made between water loss during the cuticular phase (evaporative water loss) and the respiratory as a metric to evaluate the water-saving hypothesis of DGC. For objective 2 (what is the role of the DGC in chthonic conditions?), a study needed to investigate the DGC in normal and either hypoxia/hyperoxia/hypercapnia conditions and report the metabolic rate and some form of variance. Where the flutter (F) and closed (C) phases were analyzed separately, the data were combined to generate the interburst.

For objective 3 (how much does the metabolic rate change with a degree change in temperature for insects during the DGC?), a study needed to report the mean metabolic rate, at a minimum, across two experimental temperatures under normoxia. Where applicable, mean metabolic rates and standard errors (SE) were converted to ml g^−1^ h^−1^ and standard deviations (SD), respectively, to allow for comparisons across studies. Metabolic rate data presented as ml h^−1^ were converted to ml g^−1^ h^−1^ by dividing the mean metabolic rate by the mean body mass of insects (grams), while estimates (SE/SD) were generated using a Taylor series expression (see below):$$\mathrm{Taylor}\;\mathrm{series}\;\mathrm{expression}:\;\mathrm{SE}=\frac{\mathrm{Mean}\;A}{\mathrm{Mean}\;B}\sqrt{\frac{V_{A}}{{\left(\mathrm{Mean}\;A\right)}^{2}}+\frac{V_{A}}{{\left(\mathrm{Mean}\;B\right)}^{2}}}$$
where SE = standard error, Mean A = metabolic rate at ml h^−1^, Mean B = body mass of insects in grams, and V_A_ and V_B_ = variance at A and B, respectively.

Similarly, water loss data were converted to mg h^−1^ to facilitate comparisons across all studies. All data were extracted independently by two researchers (S.O.O. and K.O.O.—see acknowledgments). Data from figures were extracted in R [27] using the *metaDigitize* package (1.0.0). When dovetailing studies provided insufficient data for inclusion in the meta-analysis (five studies), the corresponding authors were contacted via email for the possibility of providing data; however, only one corresponding author responded, and no additional data were sent.

### 2.3. Statistical Analyses

Meta-analysis was conducted in R using the *metafor* (2.1–0) and *meta* (4.11–0) packages. The random-effects model was preferred to a fixed-effect model because of the variability in experimental parameters between studies. The type of data provided in selected studies always influences the choice of effect size. Such data must be computable right from the study and should be easy to interpret. Here, because of the availability of categorical sets of data (i.e., means of groups), variances (i.e., standard deviations/standard errors), and sampling distribution supplied in each study, the Hedges’ *g* effect size metric was pertinent. Statistically, the Hedges’ *g* is a standardized mean difference that has the same meaning regardless of the study design. Therefore, we can compute the effect size and variance from each study using the appropriate formula, and then include all studies in the same analysis. For objective 1, the Hedges’ *g* effect size was used to compare cuticular and respiratory water loss (mg h^−1^). For objective 2, metabolic rates (CO_2_: ml g^−1^ h^−1^) were compared between normoxia and either hypoxia/hyperoxia using Hedges’ *g* effect size. For objective 3, the effect size from each study was calculated as a function of change (slope; $\beta_{1})\;$in respiration per °C increase in temperature. Since studies reported the mean metabolic rate (ml CO_2_ g^−1^ h^−1^) and standard error/deviation of mean rates (σ_M_), the standard error of the log-linear model (σ_LM_) was first calculated using the delta method [28,29]:
σ_LM =_ σ_M_ · mean^−1^


Then, σ_LM_ was used to calculate the sampling variance of the log-linear model slope:$$SE^{2}=\frac{(\sigma^{2}{{}_{1}}_{\;}+\sigma^{2}{}_{2})/\left(\sum \;n-2\right)}{\sum (n\;\cdot \;\left(x-`x{)}^{2}\right)}$$
where $\sigma^{2}{{}_{1}}_{\;}$is $\sum \left(\left(n\;-1\right)\;\cdot {\;\sigma }^{2}{}_{LM}\right),{\;\sigma }^{2}{}_{2}$ is $\sum \left(n\cdot {\left(\ln R-\ln f\right)}^{2}\right),$ n is the number of individuals for each mean metabolic rate at a given temperature x, and $\bar{x}$ is ∑(n · x)/∑n. lnR is the natural logarithm of respiration rates and lnf is the fitted values of the log-linear model.

The effect size from each study was calculated as a function of change in respiration per °C increase in temperature using the following equation:$$\mathrm{ES}=(e^{\beta_{1}}-1)\;\cdot \;100$$

Variance (V) in effect size was calculated using the delta method:$$V={\left(100\right)}^{2}\cdot \;e^{2\beta_{1}}\;\cdot {\left(SE\right)}^{2}\;\;$$

Since comparisons were made between respiratory water loss and cuticular water loss (objective 1), a negative effect size estimate would support the conservatory role of the DGC, while a positive effect size estimate would support otherwise. Additionally, a negative effect size estimate between either normoxia vs. hypoxia or normoxia vs. hyperoxia (objective 2) would suggest that the DGC serves to facilitate gaseous exchange during chthonic conditions, while a positive effect size suggests no correlation between the DGC and chthonic conditions.

Phylogenetic meta-analyses were completed using the *MCMCglmm* [30,31] and *ape* [32] packages in R studio version 3.6.1. [27]. The *MCMCglmm* was used to create a generalized linear mixed model. *ape* was used for reading, writing, and plotting the phylogenetic tree. Sensitivity analyses were conducted by removing effect sizes that showed negative percent changes. For all objectives, we ran a subgroup analysis by order. The underlying assumptions are that studies within each subgroup (order) do not share a common effect size and that true between-studies variance (*T^2^*) is not the same for all subgroups. Thus, *T^2^* within each subgroup was computed separately. Forest and funnel plots were either drawn in GraphPad prism (8.4.0), R studio version 3.6.1, or RevMan 5.3 [33]. Publication bias was assessed with funnel plot asymmetry and Egger’s regression test [34]. A biased dataset would be asymmetrical (i.e., skewed), while an unbiased dataset would be symmetrical [34]. In other words, the distribution of data points would be relatively even on either side of the plot. The concept of bias here draws on the adequacy of the sample size in making a reliable precision about the effect size estimate.

## 3. Results

### 3.1. Included Studies

The search strategy yielded > 1500 studies on Web of Science and PubMed, respectively (Figure 2). The title and abstract for each paper (in each search engine) were screened for relevancy and 979 papers were downloaded from Web of Science (670) and PubMed (309). After reading each article’s abstract, 179 duplicates and 569 papers that were not DGC- or DGE-specific were removed. Out of 231 papers left, 32 papers satisfied the inclusion criteria for at least one objective (see above for a list of objectives). The characteristics of the included studies are summarized in Appendix A (hereafter referred to as S). Overall, 46 insect species in 24 families in nine orders are represented in the meta-analysis (Obj. 1: five families, three orders; Obj. 2: six families, four orders; and Obj.3: 13 families, nine orders).

### 3.2. Objectives

#### 3.2.1. Objective 1: Does the DGC Reduce Water Loss in Insects?

Objective 1 incorporated seven studies with 42 sets of effect sizes between respiratory and cuticular water loss (mg h^−1^) measured during the discontinuous gas exchange cycle (DGC) in insects. Among these studies, 17 species of six families in three orders (Blattodea, Coleoptera, and Orthoptera) were represented. Although there were studies on hymenopterans that estimated water loss, there were no comparable data based on this study inclusion criterion to rationalize inclusion into this meta-analysis. To reiterate, a study needed to report mean cuticular and respiratory water loss and some form of measurement variance (SD or SE) to satisfy inclusion. Our result showed that the DGC significantly reduces respiratory water loss (−3.27 ± 0.88; estimate ± 95% confidence interval [95% CI]; *p* < 0.0001) in insects (Appendix A). Pooled effect sizes (Hedges’ g) ranged from −4.15 to −2.38. When outliers were identified and removed, the DGC was still shown to reduce respiratory water loss (−3.80 ± 0.54; estimate ± 95% CI; *p* < 0.0001, I^2^ = 38.4%) (Appendix A). Heterogeneity or between-studies variance in the model was high (I^2^ = 91%). To explain heterogeneity, a subgroup analysis was conducted, and between studies, variation was 43%, 43%, and 92% for Blattodea, Coleoptera, and Orthoptera, respectively (Figure 3). The subgroup analyses showed that the DGC is extremely important for Coleoptera (*p* = 0.02) and Orthoptera (*p* = 0.01) compared with Blattodea (*p* = 0.19). Publication bias was estimated using the funnel plot, Egger’s test, and the trim-and-fill method. Funnel plot showed a slight skewness of data to the left, while Egger’s test (intercept = −3.92; confidence interval = −1.37; t = −5.95; *p* < 0.05) showed bias, suggesting negative results may be under-reported (Figure 4).

#### 3.2.2. Objective 2: What Is the Role of the DGC under Chthonic Conditions?

Does the DGC facilitate a gaseous exchange under hyperoxia and hypoxia? This question was asked in studies where either normoxia (~21% O_2_) vs. hyperoxia (~41% O_2_) or normoxia vs. hypoxia (~10% O_2_) was investigated in insects exhibiting a DGC. The normoxia vs. hyperoxia questions was found in seven studies, with 29 effect sizes distributed in seven species in six families of four orders. Overall, the DGC was not maintained under hyperoxia (0.21 ± 0.25; estimate ± 95% CI; *p* = 0.12) (Appendix A). The between-study variance was low (I^2^ = 0%). However, a subgroup analysis (by order) indicated that the DGC facilitated gaseous exchange in dipterans (0.43 ± 0.34; estimate ± 95% CI; *p* = 0.01), but not in blattoids (*p* = 0.89) or orthopterans (*p* = 0.57) (Figure 5a).

Similarly, the DGC does not facilitate gaseous exchange in hypoxia (Z = 2.05; 0.27 ± 0.26; estimate ± 95% CI; *p* = 0.04); rather, it favors normoxia in insects (Appendix A). This estimate had a moderately low heterogeneity (I^2^ = 18%). Subgroup analysis suggested that during normoxia, the DGC was maintained in dipterans (0.34 ± 0.35; estimate ± 95% CI; *p* = 0.05) and blattoids (1.45 ± 01.26; estimate ± 95% CI; *p* = 0.02) (Figure 5b). Funnel plot analysis showed fairly even symmetry in hyperoxia and hypoxia studies (Figure 6).

#### 3.2.3. Objective 3. How Does the Metabolic Rate Change with Respect to Temperature?

How does the metabolic rate change with respect to temperature? This was estimated in 18 studies with 30 effect sizes distributed in 23 species in 12 families of nine orders, estimating the metabolic rate across a minimum of two temperatures for a single species exhibiting a DGC. The effect size (slope) of the log-linear model gives an index of the percent change in the metabolic rate per °C increase in temperature [29]. With the inclusion of order phylogeny, family, and species as random effects, the model reflected that the metabolic rate exhibited a significant, non-zero increase of 8.13% (± 3.48% 95% CI; *p* < 0.001) per °C increase in temperature (Appendix A). Order-level relatedness with corresponding meta-analytical means and 95% CIs are shown in Figure 7.

## 4. Discussion

It was the early description of discontinuous gas exchange cycle in lepidopterous insects (particularly pupae) that underpinned the roles played by spiracles and the tracheal system as the sites for modulating the release of CO_2_ [14,17,35]. For example, a significant proportion of 90% of metabolic CO_2_ accumulated within *Cecropia* pupae is expelled through the spiracles when they open briefly, and the rest is lost through the cuticle when the spiracles are closed [36]. Further studies led to the conclusion that the regulated opening and closure of the spiracles also reduces respiratory water loss (i.e., the hygric hypothesis; [7,17]) and enhances gaseous exchange in hyperoxia/hypoxia environments (i.e., the chthonic hypothesis; [16]). As advances in technology appeared—from a shift in manometric technique, electronic microbalance, and mass loss technique to flow-through respirometry—and more insect orders were investigated, hygric and chthonic hypotheses were questioned [9,37,38], and competing hypotheses arose [22,23]. As pointed out by Marias et al. [5], an idiosyncratic feature of the experiments from which competing hypotheses arose is that these studies are based on “small-scale manipulative experiments and closely related species”. Hence, comparison across multi-order levels is pertinent to evaluate the broader merit of these hypotheses. Consequently, the goal of this meta-analysis was to evaluate hypotheses by accounting for the outlier effect and weighing the findings from each study to understand the dominating or prevailing role of the DGC across insect orders. After all, insects are uniquely different in lifestyle, biology, living habitats, etc., so the DGC roles will likely vary among species, families, or orders.

The first question asked by this meta-analytic study was “Does the DGC reduce (respiratory) water loss in insects?” The focus here was the direct comparison of respiratory water loss to cuticular water loss. The meta-result provided strong support that the DGC serves to reduce respiratory water loss in insects, especially in the orders Coleoptera and Orthoptera. This is interesting and informative because this is a result obtained from another broad-scale evaluation of the hygric hypothesis. Although not a meta-analytic evaluation, White et al. [39] performed the first broad-scale phylogenetic experimental evaluation on the veracity of the hygric hypothesis. The authors found strong support for the water conservatory role of the DGC. Mechanistically, just before the burst phase, there is a build-up of CO_2_ in the tracheal. Once the CO_2_ reaches its maximum critical level, the spiracles open to allow for gaseous exchange with the environment [6,40]. A closer look at this process lends further credence to suggest that spiracular closure for a prolonged period is most likely adaptive to prevent water vapor loss [41].

Consequently, the question is why is the hygric hypothesis of the DGC unsupported by data from some few-species studies and/or insects inhabiting dry environments? A closer look at the effect size from each study from the overall forest plot (S2) showed that 74% of the effect sizes agree with the hygric hypotheses. Meanwhile, the subgroup analysis (Figure 3) showed skewness in the available literature for insect orders. Intriguingly, when a modest random literature search (i.e., looking through random articles on PubMed and Web of Science) was conducted on hypotheses refuting the hygric hypothesis, most were on studies on hymenopterans—there were no comparable data based on the inclusion criterion of this study to rationalize inclusion into this meta-analysis. On the one hand, Lighton and Turner [42] observed the correlation of events occurring in ants during DGC and DGC abolishment and outlined that the abolishment of the DGC does not influence water loss rates in ants. After all, water loss through the cuticle predominates total water loss in hymenopterans [38]. Moreover, cuticular water loss occurs during the interburst phase, which makes up more than 75% of a given ant’s DGC [43], and an ant’s cuticle is characterized by extremely low cuticular permeability, low spiracular conductance, and extremely low respiratory water loss rates [43,44]. On the other hand, Zachariassen [45] argued that even such a low water loss rate is an important “avoidable” cost to insects adapted to dry environments. Or perhaps, they have other “easy” ways to deal with water loss. Finally, there is another important question that no study, to the best of our knowledge, addresses; notably, how often does an insect exhibit a DGC during the day? Yes, the DGC serves to reduce respiratory water loss, but if an individual only does it a few minutes a day, how relevant would that be?

Beyond statistical bootstrapping, why does the hygric hypothesis fail for xeric insects and/or insects with low cuticular permeability? Perhaps this hypothesis fails because of the nature of the question being addressed and the simplistic view with which the objective is viewed. For example, arguments such as the abandonment of the DGC in conditions where water loss restriction is pertinent [46,47,48] and the insignificant proportion of respiratory water loss to total water loss [18,41,49,50] is sometimes used to discredit the water conservatory role of the DGC. In arguments like these, what is sometimes not considered, as Chown [50] put it, is the absence of the null hypothesis on what the proportion of respiratory water loss to total water loss should be? Now, even if respiratory water loss contributes a proportion to the total water loss of an insect, by coordinating the spiracles, the insect stands the chance to arguably minimize this trans-spiracular water loss rates [51]. The cuticular water loss modulation may be beyond such an insect. Of course, this is not absolute. For example, the American cockroach, *Periplaneta americana* L., can rapidly reduce its cuticular water loss [52]. Interestingly, the DGC reduces water loss in comparison to other gas patterns, corroborating the assertion that the DGC is likely maintained to minimize “avoidable” water loss for xeric insects and/or insects with low cuticular permeability. Compared to mesic insects, xeric insects have cuticular permeabilities in half the range of those recorded in the former [49,53]. Conversely, mesic insects would lose water more rapidly than xeric insects. Thus, any physiological or behavioral mechanism to minimize respiratory water loss is likely to be adaptive to xeric insects, even if such an act is not entirely consistent for mesic insects. This is probably why ants and other xeric inhabitants would probably still exhibit a DGC. Moreover, insects adapted to different environments will show a remarked difference in their ability to tolerate (desiccation tolerance) and resist (desiccation resistance) water loss [50]. Taken together, these observations caution against discrediting the water conservatory role of the DGC as not a water-saving mechanism. More importantly, it serves as a guide to interpreting the role of the DGC for any given insect species, as this is likely to go beyond spiracular closure and openness, but correlated with body mass, habitat characteristics, cuticular permeability, insect taxa, and metabolic rate [45,50,54].

The second question asked was “Does the DGC facilitate a gaseous exchange under hyperoxia or hypoxia?” In other words, is it safe to conclude that the DGC facilitates gaseous exchange in chthonic environments? The meta-result provided no support for this hypothesis; rather, it suggested that the DGC is only maintained during normoxia. This conclusion is unsurprising given the characteristics of the studies included in this objective. The selected articles included research conducted on mostly pupa and adult stages of insects. To establish the adaptive significance of spiracular closure during the respiratory gaseous exchange in insects, Schneiderman [55] noted that oxygen enters the trachea at many times the rate of carbon dioxide (due to simple diffusion: the concentration of O_2_ in the air is 20.95%, whereas the concentration of CO_2_ is 0.04%) when the spiracles are closed in *Cecropia* pupae. Hence, both periods of spiracular closure and opening will offer little resistance to oxygen entry, if any, during hyperoxia (high oxygen) or hypoxia (low oxygen relative to normoxic conditions). Similarly, the argument is that diffusion of CO_2_ away from the insect body can only occur if there is a diffusion gradient between “neat air” and “expelled air”. Therefore, insects “hold their breath” and build up a high concentration of CO_2_. However, when the spiracles open, the CO_2_ escapes from the body. Moreover, the chthonic hypotheses have been demonstrated not to lower the ratio of respiratory water loss to CO_2_ release [37,56,57]. In short, as suggested by this meta-result, the DGC might have no role in supporting the chthonic hypothesis.

Therefore, does the possibility exist that the hygric and chthonic hypotheses are mutually exclusive, or can the DGC serve two or more adaptive functions at the same time? Based on the available evidence, the DGC can, potentially, serve more than one adaptive function. Whether or not these functions can occur at the same time remains unclear. For example, Schilman et al. [23] recorded a peak in respiratory water loss after ants were placed in anoxia conditions. Similarly, a substantial increase in water loss occurs after hypercapnia was used to induce a spiracular opening [43,49,50]. Within the same colony, ant castes may exhibit different gas patterns depending on the habitat characteristics and caste roles [16,51,58]. For example, queen ants are reserved in underground chambers that are likely to have a low O_2_/high CO_2_ influx. The worker ants are not as restricted, constantly moving between the underground chamber and the outer surface (normoxia) for colony duties. In this type of scenario, Lighton and Berrigan [16] noted that the gas patterns were remarkably different between queens and workers. Hence, for the queen, the DGC is most likely employed during anoxia to “firstly” facilitate gaseous exchange before “secondly” minimizing respiratory water loss [59]. For the worker that forages, the DGC would most likely be employed to reduce transpirational water loss rates [51] given external conditions. One cannot but wonder, could multiple “small adaptations” lead to or reinforce the DGC?

The third question asked by this meta-analytic study was “How much does the metabolic rate change with a degree change in temperature (°C) for insects breathing during the DGC?” Insect metabolic rates can be affected by several factors, including temperature, reproduction, and feeding (Waters and Harrison, 2012; Henrich and Bradley, 2014), but no attempt has been made to scale how metabolic rate changes per unit increase in temperature for insects during the DGC. Mechanistically, the DGC is initiated by the interburst phase, where uptake of O_2_ occurs in endotracheal cells with simultaneous catabolic production of CO_2_ that accumulates in the hemolymph. O_2_ pressure in the endotracheal system reaches a critical setpoint and the insect’s spiracular muscles become inactivated due to CO_2_ build-up to allow for air outflow (i.e., burst phase) [6,43]. Thus, a scaling metabolic rate (${\dot{V}}_{{\mathrm{CO}}_{2}}$) with temperature may explain how O_2_ uptake and CO_2_ emission in insects change in response to temperature [60]. Such information can explain how the metabolic rate scales to thermal sensitivity [61], how the DGC controls the rate at which an insect transforms energy and materials [62], and how temperature influences the rate of CO_2_ emission in insects. We made this scaling using slope instead of temperature coefficient (i.e., *Q*_10_) values. The comparison of studies using slope is advantageous over common *Q*_10_ values in two ways. It can be used to compare metabolic rate across more than two temperatures and its interpretation does not require reference to other *Q*_10_ values [29]. Irlich et al. [61] conducted a meta-analytic evaluation of metabolic-rate temperature relationships on a global level (i.e., irrespective of the gas pattern) in insects. Effect sizes were calculated from 37 families distributed in nine orders. Like this meta-analytic study, Irlich et al. [61] utilized the slope of the metabolic rate temperature but described their results in terms of mean activation energy of the respiratory complex (0.62 eV). Activation energy is an index of temperature dependence term of the metabolic theory of ecology [62]. This study estimated effect sizes from 18 studies, with 30 effect sizes distributed in 23 species in 12 families of nine orders. The meta-result in this study indicates that metabolic rate exhibits a significant, non-zero increase of 8.13% per °C (a *Q*_10_ value of 2.02) increase in temperature during the DGC. Further understanding is required to establish the link between mean activation energy and slope. It may also be informative to consider the 8.13% per °C increase in temperature in the context of ecological implications. Schilman et al. [23] discussed how scaling the metabolic rate with the temperature rate of an insect can be factored into mathematical models to predict the vectorial capacity in propagating diseases. For example, an increase in the metabolic rate could drive catabolism in insects, thereby accelerating the rate of feeding and development. Such an occurrence would increase the burden of agricultural pest insects on crops. Even so, using ${\dot{V}}_{CO_{2}}$ as a proxy for the metabolic rate across species has its limitations. First, ${\dot{V}}_{CO_{2}}$ is less accurately translated to energy metabolism units and subject to give false signals in the presence of an acid-base imbalance [63]. Moreover, the energy equivalence of ${\dot{V}}_{CO_{2}}$ varies with respiratory quotient (RQ). Interestingly, RQ can also vary with temperature and from one species to another [64].

It is known that an increase in temperature can trigger a shift in gas exchange patterns [9,65]. However, that is not the case here, as all data were retrieved from insects that breathe discontinuously over acute temperatures (i.e., a range of temperatures). As highlighted by Terblanche et al. [66], an acute increase in temperature is directly proportional to the metabolic rate and a given DGC frequency. Thus, one can assume that such a change in the cycle frequency is likely to hold important implications for water balance. While an inverse relationship between the metabolic rate and temperature is well documented in insects across all gas patterns (see [66] and references therein), our results show that metabolic rate exhibits an 8.13% per °C increase in temperature, at least for insects breathing discontinuously. Although not directly related, another meta-analytic study found that copepods respiration increases by 7% per °C increase in temperature [29]. Thus, we suggest that this range reflects the general characteristics of arthropod ectothermic poikilotherms. Considering that the DGC is mostly exhibited by quiescent insects, and an increase in metabolic rates predates water loss [58,67], further clarity needs to be sought as to what the adaptive significance and or implication of this metabolic rate increase is to insects. On the one hand, Terblanche et al. [66] evinced that water loss rates were reduced in response to the acclimatization of high temperatures. On the other hand, our result here may inflate that observation. Understandably, the work by Terblanche and colleagues [66] was conducted on a single dung beetle species, while this is a realization from broad-scale studies.

Finally, and this goes for all the adaptive hypotheses suggested to explain the significance of DGC, the concept of adaptation, even though suggested/mentioned, is often overlooked. This is not the case within the Darwinian concept. The Darwinian concept of adaptive-*ism* argues that these traits must give or make the insect better able to survive and reproduce compared to others that lack those traits. Importantly, these traits must be heritable [68]. First, no study has been conducted to investigate the genetic relevance (basis) of gas pattern respiration. After all, for it to be adaptive, there must be a genetic basis. Second, to the best of our knowledge, no study exists comparing the relevance of the DGC on a large scale to the biological fitness of insects (of course, this would need to be defined using a heuristic approach). Studies involving *Drosophila melanogaster* Meigen have demonstrated the capacity of desiccation-resistant populations to evolve and recover from the effects of desiccation at a rate more than non-desiccant-resistant populations [69]. By extension, such a distinct advantage should be sought for insects that make use of the DGC and those that do not.

It is noteworthy that the three questions asked in this meta-analytic study combined data from insects with varying life stages, body masses, treatment types, number of spiracles, and habitats (S1). While the model for the third question accounted for these nuances, we exercise caution in interpreting these data as absolute. We have only presented a holistic approach to solving the significance of the adaptive hypotheses posited to explain DGC occurrence in insects. Understandably, there are a few ways to circumvent these inherent variabilities—all of which would require data that are lacking. One way would be to design small-scale experiments involving the same technique/protocol and environmental parameters for each insect order/group of species. Thereafter, a meta-analysis can be sought.

## 5. Conclusions

As pointed out earlier, the skewness of the available data in this meta-analytic study to include nine out of the possible 31 insect orders may represent one major limitation to the interpretations from this study. To compare across a phylogenetic broad-scale study, Marias et al. [5] and White et al. [39] had to provide new information by conducting experiments of orders unavailable in the literature. Such an approach is beyond the scope of any meta-analytic study. It is possible that upon the availability of data from other orders, the conclusions may change. This possibility remains yet unseen, and on the premise of the available literature, the meta-result indicates three conclusions; (1) DGC serves to minimize respiratory water loss, (2) DGC does not facilitate gaseous exchange in hyperoxia/hypoxia, and (3) the metabolic rate exhibits a significant, non-zero increase of 8.13% per °C increase in temperature during DGC. These data represent the first quantitative meta-analysis attempt to resolve the controversies surrounding the merit of adaptive hypotheses in insects.

## Figures and Tables

**Figure 1 insects-13-00117-f001:**
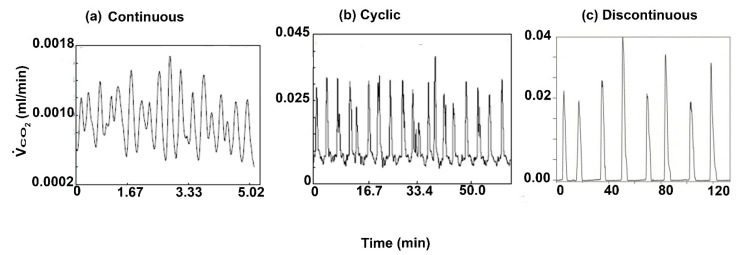
An illustration of the consensus of the three types of gas patterns commonly observed in insects: (**a**) continuous [5], (**b**) cyclic [5], and (**c**) discontinuous gas exchange [9].

**Figure 2 insects-13-00117-f002:**
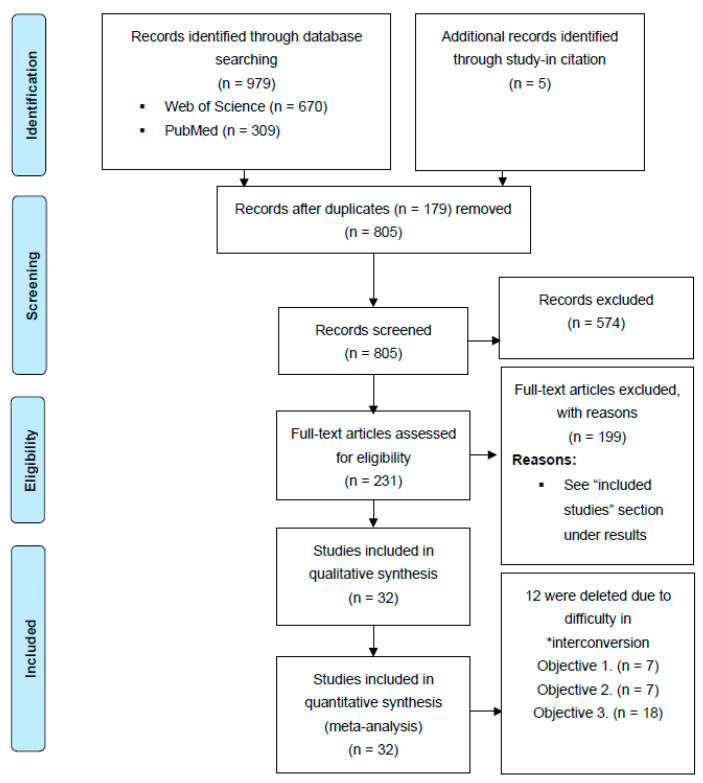
PRISMA flow diagram of study identification, eligibility screening, and inclusion. * interconversion = the inability to convert units reported in those studies to units in this meta-analysis due to lack of requisite data (see “*inclusion criteria and data extraction”* section).

**Figure 3 insects-13-00117-f003:**
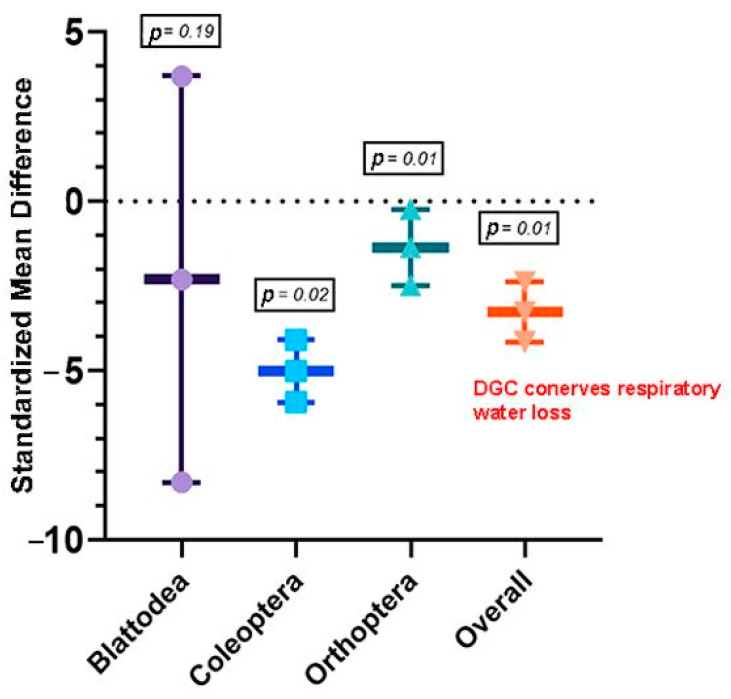
Meta-analytical, subgroup estimate of water-saving hypothesis evaluation during the DGC with 95% CI in Blattodea, Coleoptera, and Orthoptera insect orders. The overall effect represents the collective effect of all three orders. Effect size estimates (measured as the standardized mean difference; Hedges’ *g*) with 95% CIs that do not overlap zero are considered statistically significant at *p* = 0.05. *p*-values are provided in the textbox.

**Figure 4 insects-13-00117-f004:**
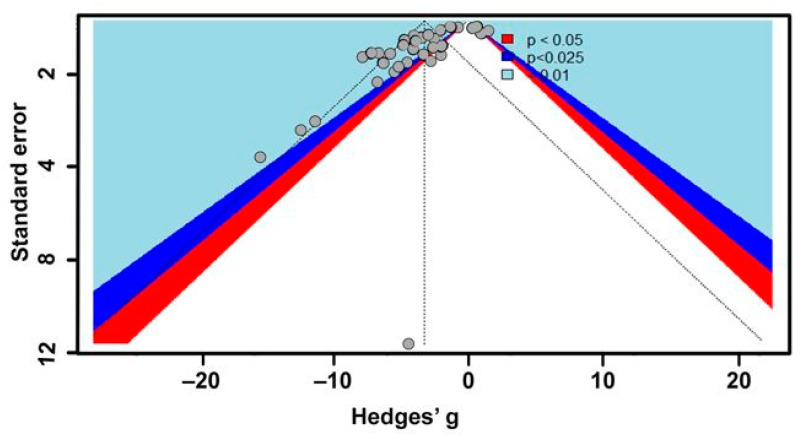
Funnel plot for water-saving hypothesis studies (Objective 1: Does the DGC reduce water loss in insects?). The light blue, dark blue, and red areas correspond to 99%, 99.75%, and 95% confidence intervals, respectively. Asymmetric distribution of studies indicates publication bias.

**Figure 5 insects-13-00117-f005:**
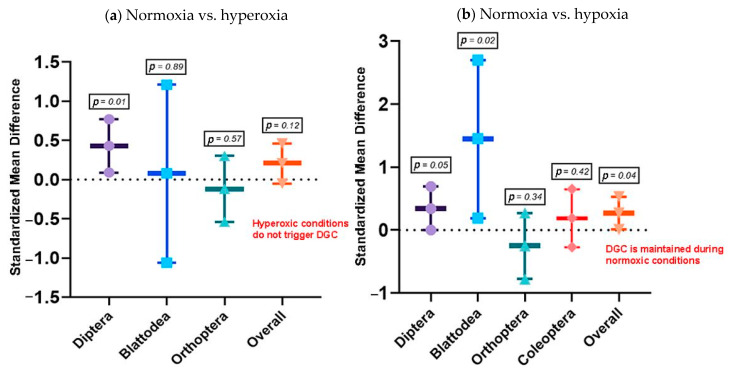
Meta-analytical, subgroup estimate of chthonic hypothesis in (**a**) normoxia vs. hyperoxia, and (**b**) normoxia vs. hypoxia conditions in insects during the DGC with 95% CI. The overall effect represents the collective effect of all three orders. Effect size estimates with CIs that do not overlap zero are considered statistically significant at *p* = 0.05. *p*-values are provided in the textbox.

**Figure 6 insects-13-00117-f006:**
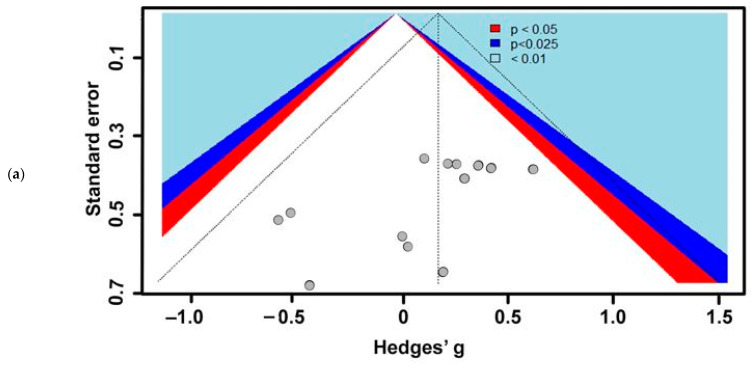
Funnel plot of (**a**) normoxia–hyperoxia and (**b**) normoxia–hypoxia discontinuous gas exchange studies. The dotted line represents a 95% confidence interval. The slightly skewed distribution of studies suggests a possible publication bias.

**Figure 7 insects-13-00117-f007:**
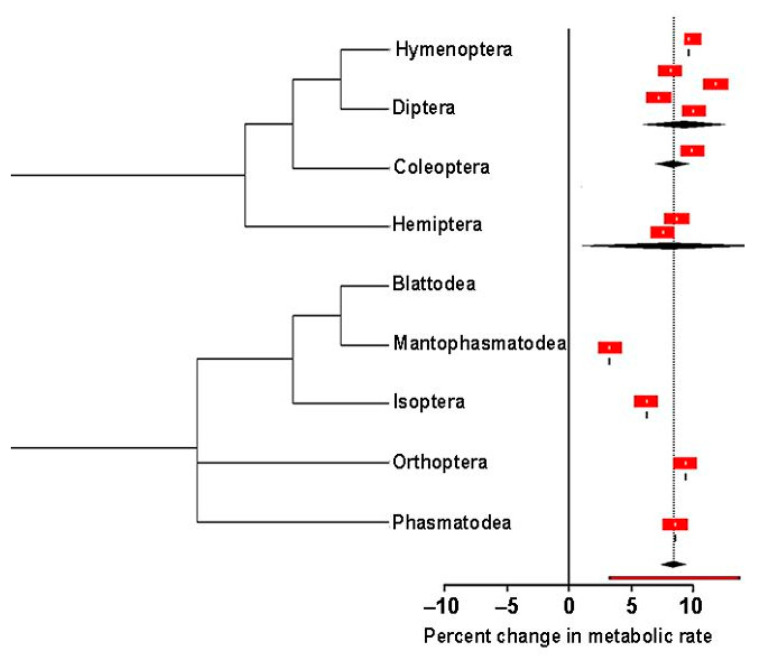
Meta-analytical, mean percent changes in metabolic rate of insects respiring through the
DGC with 95% CIs of insect orders. Effect size estimates with CIs that do not overlap zero are considered
statistically significant (*p* = 0.05). The phylogenetic tree is redrawn from Gullan and
Cranston, 2000.

## Data Availability

All code used to analyze data in this manuscript is available at: https://doi.org/10.5061/dryad.ht76hdrdz (accessed on 18 January 2022).

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
