# Peer review of "Why Do Insects Close Their Spiracles? A Meta-Analytic Evaluation of the Adaptive Hypothesis of Discontinuous Gas Exchange in Insects"

_insects, 2022, doi:10.3390/insects13020117_

Round 1

Reviewer 1 Report

The manuscript presents a particular approach for attempting to shed light on a long-lasting unsolved question in insect physiology, i.e. the functional significance of discontinuous respiration. Different approaches have been employed in order to discriminate between different hypotheses and being the object of recent publications. The authors conducted a metanalysis of a significant amount of published studies and provide their results and conclusions. The study is interesting and timely, yet some passages appear to be overly declamatory, rather than factual. Some concepts and terms should be presented in a more precise form.

  • Lines 19-21: “…synthesizing data from across all insects where discontinuous gas exchange cycle has been reported.” This claim seems to exaggerate the number of datasets actually treated in this study. Please, clarify.
  • The statement in lines 42-43 about the role of the circulatory system in insect respiration is incorrect. It can be only understood if authors are referring to circulating respiratory pigments exclusively, but this is not specified. Indeed, hemolymph circulation plays a major role in tracheal ventilation in many insects, probably most of them. Please, correct.
  • Lines 52-53 make the reader believe that the ancient notion that “gas exchange is always by diffusion” would be true, even though it has been discarded a long time ago. Please, clarify.
  • Line 67: what is “To understand this, it is important to demystify the gas patterns in insects.” supposed to mean? Please, clarify or delete the sentence.
  • Lines 96 and following. This section is confusing, and the relevance of key factors, as the variation of the partial pressure of each gas, the retention of carbon dioxide in tissues, and the variation in the internal pressure of the insect body should be mentioned.
  • PRISMA statement does not need to be explained in detail. A succinct presentation of the general concept plus a couple of adequate references would suffice.
  • Lines 141-162. Please, better justify these allegations. I can understand that metanalyses could provide additional certitude by integrating results of other approaches, but only if they are based on original complete datasets or complete descriptive statistics of each study. Is this always the case?
  • Stating that metanalyses are “less biased than conventional approaches” needs further explanation. Please, indicate the approaches and their associated biases. In particular, because statements in lines 598-600 of the Conclusions, explicitly admit that metanalyses may be biased since a wide covering is “beyond the scope of any meta-analytic study”, which is in contradiction with the previous claims of their power and objectivity over other approaches. Please, clarify.
  • Please, clarify, using a consistent definition for the word “respiration” (cellular? ventilation? gas exchange?).
  • The abbreviation of degree Celsius is not “oC” but “°C”. Please, correct.
  • Authors claim to compare “metabolic rates” (i.e. energy expenditure rates), but they indicate using exclusively carbon dioxide emission (VCO2) data. The measure of CO2 emission is a proxy to O2 consumption, which is a proxy to the metabolic rate (MR). The latter can only be estimated if other variables are known, as the respiratory quotient. A limitation is that neither the respiratory quotient (RW= ratio VCO2/VO2), nor the ratio VO2/MR are constant, both depending on the chemical substrate, i.e. carbohydrates, lipids, or protein, being catabolized. Besides, RQ can also vary for the same species (e.g. Leis et al., J. Exp. Biol., 2016). As a result, comparing VCO2 or changes in VCO2 across species is not straightforward. This limit of the approach must be clearly stated, and terminology accurately employed.

Author Response

Comments and Suggestions for Authors

The manuscript presents a particular approach for attempting to shed light on a long-lasting unsolved question in insect physiology, i.e. the functional significance of discontinuous respiration. Different approaches have been employed in order to discriminate between different hypotheses and being the object of recent publications. The authors conducted a metanalysis of a significant amount of published studies and provide their results and conclusions. The study is interesting and timely, yet some passages appear to be overly declamatory, rather than factual. Some concepts and terms should be presented in a more precise form.

  • Lines 19-21: “…synthesizing data from across all insects where discontinuous gas exchange cycle has been reported.” This claim seems to exaggerate the number of datasets actually treated in this study. Please, clarify.

Response: We agree with the reviewer that such a phrase might be hyperbolic. Consequently, we have reworded it to reflect that “synthesizing apposite data…” The inclusion of the word “apposite” highlights that only data that fit our inclusion criteria were used

  • The statement in lines 42-43 about the role of the circulatory system in insect respiration is incorrect. It can be only understood if authors are referring to circulating respiratory pigments exclusively, but this is not specified. Indeed, hemolymph circulation plays a major role in tracheal ventilation in many insects, probably most of them. Please, correct.

Response: As recommended by the reviewer, we have deleted the incorrect statement about the role of the circulatory system in insect respiration

  • Lines 52-53 make the reader believe that the ancient notion that “gas exchange is always by diffusion” would be true, even though it has been discarded a long time ago. Please, clarify.

Response: We have included “either convection (i.e., bulk flow) and/or diffusion” to prevent          selling to an unsuspecting reader that gas exchange is only diffusive

  • Line 67: what is “To understand this, it is important to demystify the gas patterns in insects.” supposed to mean? Please, clarify or delete the sentence.

Response: Sentence deleted

  • Lines 96 and following. This section is confusing, and the relevance of key factors, as the variation of the partial pressure of each gas, the retention of carbon dioxide in tissues, and the variation in the internal pressure of the insect body should be mentioned.

Response: The goal of “Lines 96 and following” was to provide an overview of the hypotheses posited to explain the occurrence of DGC in insects and what needs to be true for each hypothesis to hold. While we don’t disagree that the reviewer’s key factors are pertinent, there is, modestly put, a dearth of studies providing quantitative data on these key factors. As such, a meta-analysis of using these key factors might be impossible

  • PRISMA statement does not need to be explained in detail. A succinct presentation of the general concept plus a couple of adequate references would suffice.

Response: True, the PRISMA statement does not need to be explained in detail. However, for the sake of readers who aren’t familiar with meta-analysis and PRISMA, we have taken the liberty to provide this information to at worst start as a guide or at best to enable reproducibility

  • Lines 141-162. Please, better justify these allegations. I can understand that metanalyses could provide additional certitude by integrating results of other approaches, but only if they are based on original complete datasets or complete descriptive statistics of each study. Is this always the case?

Response: Yes, metanalyses could provide additional certitude by integrating the results of other approaches. Importantly, meta-analysis does a nice job exploring the importance of variation within and across studies.   But to perform a metanalysis, some form of data is required at least. This could be the descriptive statistics e.g., a mean and some form of variance. The complete datasets are not required.

  • Stating that metanalyses are “less biased than conventional approaches” needs further explanation. Please, indicate the approaches and their associated biases. In particular, because statements in lines 598-600 of the Conclusions, explicitly admit that metanalyses may be biased since a wide covering is “beyond the scope of any meta-analytic study”, which is in contradiction with the previous claims of their power and objectivity over other approaches. Please, clarify.

Response: Metanalysis in the strict statistical sense isn’t biased, It is the data that fit into them that can be skewed.  For example, the metanalytic calculation of effect sizes (estimating the magnitude of interest with respect to the study sample size) gives it more power by reducing the probability of Type II error. The skewness discussed in Lines 598-600 is about the type of data that we retrieved from the literature. It is an indictment of the skewness of the investigations being sought. For example, we could only find published data for nine out of 31 insect orders. Marias et al (2005) had to conduct additional experiments to obtain DGC data lacking for several insect orders. It is the conduction of additional investigations that is beyond the scope of any meta-analytic study. Metanalysis involves asking questions using available (already published) data not generating new data

  • Please, clarify, using a consistent definition for the word “respiration” (cellular? ventilation? gas exchange?).
  • The abbreviation of degree Celsius is not “oC” but “°C”. Please, correct.

Response: We thank the reviewer for spotting the error. oC” has been changed to “°C”. Kindly see line 605

  • Authors claim to compare “metabolic rates” (i.e., energy expenditure rates), but they indicate using exclusively carbon dioxide emission (VCO2) data. The measure of CO2emission is a proxy to O2 consumption, which is a proxy to the metabolic rate (MR). The latter can only be estimated if other variables are known, as the respiratory quotient. A limitation is that neither the respiratory quotient (RW= ratio VCO2/VO2), nor the ratio VO2/MR are constant, both depending on the chemical substrate, i.e. carbohydrates, lipids, or protein, being catabolized. Besides, RQ can also vary for the same species (e.g. Leis et al., J. Exp. Biol., 2016). As a result, comparing VCO2 or changes in VCO2 across species is not straightforward. This limit of the approach must be clearly stated, and terminology accurately employed.

Response: We agree with the reviewer, comparing VCO2 as a proxy to metabolic rate is not without its limits. Even so, it is, arguably, the metric most obtainable through flow-through respirometry. This is evidenced by the number of authors reporting this metric. Nevertheless, we have discussed this limitation. Kindly see lines 561-565

Reviewer 2 Report

Dear authors,

This is a good and particular interesting paper but while reading it and cross-checking other papers on this topic, I was on two minds about the title, the content and the results of this paper which - in my opinion - has a somewhat misleading title and made me expect something differently: some kind of hypothesis driven research rather than an analysis which is hard to understand.

This may be due to the fact that I looked on this DGC stuff from more a physiological and morphological background/perspective which may make me look differently at the questions and answers you asked and got from your work.

I especially liked the discussion very much. A lot of controversial aspects which make it really hard for everyone to find a simple question in this complex topic (the reason why you wrote the paper) had already been marked by myself. Please consider to add some of this information in the introduction section.

Finally, reading the discussion section, I liked this paper very much but have some reservations. I could not always follow up with the reasons why the authors used the factors they summarized in the attached Excel-sheet. Factors like maximum carbon dioxide release rate and e.g. interburst duration is in my opinion one critical parameter which will influence overall water loss on a very important way. I would suggest that the authors use more of these available date in their current or a future study.

I would suggest to include a small graphical schematic drawing which parameters you used for you evaluation. Also, at the moment I am not sure if such a small database with so many variables like insect stage (pupae, larva, adults), lifestyle (routinely flying, walking, resting), body mass, spiracle numbers, spiracular conductance, pressure differences, abdomen movements and others are sufficient to find an answer to you initial question.

Please find my comments in the follwing table. They are sorted in the order of appearence in the text:

53

The reviewer does not fully agree with the diffusion statement of the authors. The cited paper from Woods states that "In insects that use both convective and diffusive exchange (14), fluxes occur through spiracles, which are short tubes much like those used by plant leaves and bird eggshells."

56

Unfortunately, meanwhile the CO2-concentration has reached ~0.04%

56

I would suggest to call it "partial pressure gradient" rather than concentration gradient

58

Calculation error? One order of magnitude bigger real partial pressures: The PO2 should be around 158 mmHg or 21 KPa. See line 497.

63

The fact that the authors state that there is more than one pattern of gas exchange makes the intention of the paper questionable. How do the authors make sure they use the "environment" which is comparable to other papers used?

64

"gas exchange pattern" seems a better expression, see also lines 71, 72 and following lines …

77-80

The authors probably question their statement from line 53 where they focus on diffusion. There have, however, been a few papers - including the old ones in moth pupae- where pressure differences between tracheal system and atmosphere had been detected. Therefore I agree with the statement made here but also would point to the problem of repeatability if it comes to the recording methods and the results obtained from these which - in my opinion makes it very difficult to compare the results of these papers in regard to water loss rates! I think that the term interburst used in the authors dataset is appropriate to name the phase between two opening phases.

87

It may be useful to change the term "pressure (i.e., PO2)" into "oxygen partial pressure (i.e., PO2)" because there have been recordings of hydrostatic pressure in moth pupae too.

89

Your mentioned the "flow" of oxygen into the tracheal system along an pressure gradient which in my understanding is a parameter of convective gas exchange too. See my comments on diffusion and convection above and your text in line 100.

91

"tracheal system" rather than trachea system

98

"This hypothesis is strengthened by the inflow of CO2 (and water) during the F-phase." In my understanding this is better described by the term "water and CO2 retention" because the convective inflow of air inhibits the release of CO2 and water on the opposite direction.

112

"Interestingly, for this hypothesis to be also true, the F-phase 112 has to be convectional, otherwise, the hypothesis becomes weakened." This is mostly true but in the few cases when pressure in the tracheal system was measured (the classical works, Kestler´s work, some work by Slama´s microflow measurements and experiments done in the Hetz lab) it could be shown that there was a clear convective contribution to gas exchange. In the classical papers on moth pupae the F-phase turned out to be the longest phase in the whole cycle and may thus be of greater importance for water retention. Unfortunately most of the work is restricted to measurement of CO2-exchange which cannot show whether spiracles are open or closed and whether there is a pressure difference.

114-122

I would not call it easy to compare and evaluate hypotheses. As mentioned above, without a complex system of pressure, body movement and oxygen measuring devices it is still hard to tell the phases of a gas exchange in single insects. See also your text on integrating results of experiment on line 138.

123

This is important to mention: In the reviewers opinion, the whole DGC discussion has to be divided into a mechanistic/regulation aspect discussion (how it is performed, what goes on in the regulation of spiracular activity, see emergent properties), a physiological discussion (how is physiology involved in this part of the story) and an evo/devo or adaptive discussion (e.g. the reasons and benefits behind this patterns, the significance and the differences and common features of animals which show such behavior). I do not think that the adaptive hypothesis questions can be answered from simple fitting data to a model. There have, however, been many papers on each part of the story which may help in doing a more appropriate selection of parameters to be included in such a meta analysis. I therefore doubt that a discussion which focusses only on one part of the story can help in solving the ultimate questions.

166

I personally find it somewhat hard to evaluate a single factor (by reading over it I considered many other factors), the carbon dioxide release rate (which is stated by the authors as a relevant intrinsic factor affecting metabolic rate) to be an important factor for DGC between specimens and species. The authors stated above that carbon dioxide can be buffered (see line 88) and thus does not represent the actual carbon dioxide output at steady state conditions. Therefore I doubt that carbon dioxide release rate (which has to be measured over many complete cycles in order to get a significant result) is a good parameter. Furthermore, there is no measure of conductance of the tracheal system and spiracles, tracheal volume and buffer capacity of the tissues and hemolymph given in the datasets. These may vary from "burst" to "burst" and may affect pattern generation and in turn the outcome of carbon dioxide release.

169

I doubt that the selection of files which only measure carbon dioxide release rate may help in answering these questions of the ultimate reason why insects close their spiracles.

174

The goal of the study is well defined, but in my opinion the title and the objectives are a little bit out of focus.

Figure 1

Since not everyone knows these type of recording: It would be great to use the same axes-scaling in order to highlight the variation in amplitude and frequency of these events.

Figure 2

I like this diagram

227

I can see that the authors used some prominent features of the DGC which the reviewer also regards as important. There are, however, no criteria on the magnitude of these

231

I agree with the flow rate being important in order to evaluate the time response of the system. But this also largely depending on volume of the respirometer chamber and tubing length. At least the recording devices seem uniform throughout the studies selected.

241

Good. I see that combining data into a interburst data set are an appropriate way to overcome the limitations of deciding which phase in the DGC-cycle we are talking about.

245 

In you data set the body masses spread over a relatively big range (in obj2) from ~30mg to nearly 2g. Unfortunately the body masses for obj3 are not shown but may vary by around 2 orders of magnitude too. Did you take into account the nonlinear relation between body mass and mass specific metabolic rate (µl/g min) in your further evaluation considerations? A small animal (e.g. 20mg) with higher metabolic rates may probably face the same problem as a 2g beetle having a lower mass specific metabolic rate? You could have included this into a log-lin model like the T/MR-Model.

245 and 254

Wouldn´t it have been more appropriate to convert both data on carbon dioxide emission and water loss rate into molar fluxes (e.g. µmol/g h) instead of converting one of them into a volume flow (µl/g min) and the other one into a mass flow (mg/min) and NOT converting it into a mass specific rate (mg/g min)? In my opinion this procedure could have given more insights like the ratio of water loss to carbon dioxide release rate or a transpiration vs. respiration quotient which might be a suitable measure of water saving. Please reconsider.

263

You mentioned the variability in experimental procedures. This is a critical point as I expressed above. A short section on this problem would be of big use for the introduction part of the work.

325 and fig. 3

I am not really surprised by the finding.

351

I am surprised that the DGC was not shown under hyperoxia. This may come from a limited dataset and the selection of species which do not live in habitats with low O2 and high CO2 mentioned by the original hypothesis. Also the variability and selection of either pupae, larvae and adults and a wide range of weight may affect these results. Also, I wonder whether the maximum carbon dioxide release rate rather than the average (mean?) would be a more appropriate measure to be included. I was not able to check all cited data for this relation.

387

It is hard for me to figure out what significance this result has for the initial question of the paper.

404

This is one crucial point. The role of single spiracles in the DGC which has not been asses here.

413

I would not call this a simple advance in technique. Many methods applied in the older work from Schneiderman, Beckel, Levy, Kanwisher were very important. These techniques (pressure, O2/CO2 concentration/abdomen movement) have not been developed like flow through respirometry (which has the advantage of high temporal resolution) but respirometry is calibrated on these older techniques. I would suggest a more appropriate wording.

419

One could draw the alternative conclusion that these small scale experiments should be extended into other arthropod groups involving the same techniques, methods, protocols and the same environmental parameters.

discussion

The discussion section is great which came a little bit surprising for me after I read through all the previous parts of the paper which made me think of limitations in answering the initial question of the study. It shows a lot of methodological problems (which I mentioned in parts before) and a lack of data which - in my opinion can not only be gained by flow through respirometry and/or broad scale evaluation of data from other taxa.
Especially interesting is the question for any measurable fitness parameters which could be an important evolutionary drive in eco-physiological adaptation. Comparing animals from different collection sites (with crossing experiments) could also advance the knowledge about the significance of the DGC. Testing heritability (morphologically, physiologically) would be a great experiment.

Author Response

Comments and Suggestions for Authors

Dear authors,

This is a good and particular interesting paper but while reading it and cross-checking other papers on this topic, I was on two minds about the title, the content and the results of this paper which - in my opinion - has a somewhat misleading title and made me expect something differently: some kind of hypothesis driven research rather than an analysis which is hard to understand.

This may be due to the fact that I looked on this DGC stuff from more a physiological and morphological background/perspective which may make me look differently at the questions and answers you asked and got from your work.

I especially liked the discussion very much. A lot of controversial aspects which make it really hard for everyone to find a simple question in this complex topic (the reason why you wrote the paper) had already been marked by myself. Please consider to add some of this information in the introduction section.

Finally, reading the discussion section, I liked this paper very much but have some reservations. I could not always follow up with the reasons why the authors used the factors they summarized in the attached Excel-sheet. Factors like maximum carbon dioxide release rate and e.g. interburst duration is in my opinion one critical parameter which will influence overall water loss on a very important way. I would suggest that the authors use more of these available date in their current or a future study.

I would suggest to include a small graphical schematic drawing which parameters you used for you evaluation. Also, at the moment I am not sure if such a small database with so many variables like insect stage (pupae, larva, adults), lifestyle (routinely flying, walking, resting), body mass, spiracle numbers, spiracular conductance, pressure differences, abdomen movements and others are sufficient to find an answer to you initial question.

Please find my comments in the follwing table. They are sorted in the order of appearence in the text:

53

The reviewer does not fully agree with the diffusion statement of the authors. The cited paper from Woods states that "In insects that use both convective and diffusive exchange (14), fluxes occur through spiracles, which are short tubes much like those used by plant leaves and bird eggshells."

Response: We have included “either convection (i.e., bulk flow) and/or diffusion” to prevent selling to an unsuspecting reader that gas exchange is only diffusive. Kindly see line 55

56

Unfortunately, meanwhile the CO2-concentration has reached ~0.04%

Response: Indeed, the CO2 concentration in the atmosphere has now reached 412.5 ppm (https://www.climate.gov/news-features/understanding-climate/climate-change-atmospheric-carbon-dioxide). Consequently, we have corrected this by changing 0.03 to 0.04% as suggested by the reviewer. Kindly see line 58

56

I would suggest to call it "partial pressure gradient" rather than concentration gradient

Response: We take the reviewer's suggestion. “concentration gradient” has been changed to “partial pressure gradient”. Kindly see line 58

58

Calculation error? One order of magnitude bigger real partial pressures: The PO2 should be around 158 mmHg or 21 KPa. See line 497.

Response: Yes, these were calculation errors. The calculations have been redone and the right values have been inputted. Kindly see lines 59-60

63

The fact that the authors state that there is more than one pattern of gas exchange makes the intention of the paper questionable. How do the authors make sure they use the "environment" which is comparable to other papers used?

Response: Unfortunately, there is no way we could have made sure all data used the same “environment”. If we had done so, we would have skewed the data selected thereby creating paper selection bias and file drawer problem. This is why in Lines 224-232, we explicitly noted that we created less stringent exclusion criteria. We embraced the variability in treatment types (“environment”) as a strength to help us get a global picture of what the story is across multiple species. Even with this, we pointed out in Lines 601-604, that we could only get data for niner out of possible 31 insect orders

64

"gas exchange pattern" seems a better expression, see also lines 71, 72 and following lines …

Response: “exchange” has been included in the phrase “gas pattern”. Kindly see lines 66 and 70

77-80

The authors probably question their statement from line 53 where they focus on diffusion. There have, however, been a few papers - including the old ones in moth pupae- where pressure differences between tracheal system and atmosphere had been detected. Therefore I agree with the statement made here but also would point to the problem of repeatability if it comes to the recording methods and the results obtained from these which - in my opinion makes it very difficult to compare the results of these papers in regard to water loss rates! I think that the term interburst used in the authors dataset is appropriate to name the phase between two opening phases.

Response: We agree with the reviewer’s point of view. And have since made changes to a more robust focus i.e., convective and diffusive. Kindly see line 55

87

It may be useful to change the term "pressure (i.e., PO2)" into "oxygen partial pressure (i.e., PO2)" because there have been recordings of hydrostatic pressure in moth pupae too.

Response: We take the reviewer’s suggestion and used the term “oxygen partial pressure”. Kindly see line 89

89

Your mentioned the "flow" of oxygen into the tracheal system along an pressure gradient which in my understanding is a parameter of convective gas exchange too. See my comments on diffusion and convection above and your text in line 100.

Response: We have included “either convection (i.e., bulk flow) and/or diffusion” to prevent selling to an unsuspecting reader that gas exchange is only diffusive. Kindly see line 55

91

"tracheal system" rather than trachea system

Response: “Trachea system” has been changed to “tracheal system”. Kindly see line 91

98

"This hypothesis is strengthened by the inflow of CO2 (and water) during the F-phase." In my understanding this is better described by the term "water and CO2 retention" because the convective inflow of air inhibits the release of CO2 and water on the opposite direction.

Response: We take the reviewer’s suggestion. Consequently, the “inflow of CO2 (and water)” has been changed to “Water and CO2 retention”. Kindly see line 101

112

"Interestingly, for this hypothesis to be also true, the F-phase 112 has to be convectional, otherwise, the hypothesis becomes weakened." This is mostly true but in the few cases when pressure in the tracheal system was measured (the classical works, Kestler´s work, some work by Slama´s microflow measurements and experiments done in the Hetz lab) it could be shown that there was a clear convective contribution to gas exchange. In the classical papers on moth pupae the F-phase turned out to be the longest phase in the whole cycle and may thus be of greater importance for water retention. Unfortunately, most of the work is restricted to measurement of CO2-exchange which cannot show whether spiracles are open or closed and whether there is a pressure difference.

Response: This observation has been duly noted

114-122

I would not call it easy to compare and evaluate hypotheses. As mentioned above, without a complex system of pressure, body movement and oxygen measuring devices it is still hard to tell the phases of a gas exchange in single insects. See also your text on integrating results of experiment on line 138.

Response: We agree with the reviewer. The use of the word “easily” in the sentence “Importantly, the three phases of DGC provide a mechanistic way of easily comparing and evaluating hypotheses” is a trivial way of stating it. Thus, we have deleted the word “easily” from the sentence. Kindly see line 117

123

This is important to mention: In the reviewers opinion, the whole DGC discussion has to be divided into a mechanistic/regulation aspect discussion (how it is performed, what goes on in the regulation of spiracular activity, see emergent properties), a physiological discussion (how is physiology involved in this part of the story) and an evo/devo or adaptive discussion (e.g. the reasons and benefits behind this patterns, the significance and the differences and common features of animals which show such behavior). I do not think that the adaptive hypothesis questions can be answered from simple fitting data to a model. There have, however, been many papers on each part of the story which may help in doing a more appropriate selection of parameters to be included in such a meta-analysis. I therefore doubt that a discussion which focusses only on one part of the story can help in solving the ultimate questions.

Response: The reviewer has highlighted an important point. A type that we have embraced and upon afterthought required the inclusion of a paragraph to highlight the limitations of this meta-analysis given the data used. Consequently, we have stated the nuances of the data and suggested ways to improve should a subsequent meta-analysis be desired. Kindly see lines 602-609

166

I personally find it somewhat hard to evaluate a single factor (by reading over it I considered many other factors), the carbon dioxide release rate (which is stated by the authors as a relevant intrinsic factor affecting metabolic rate) to be an important factor for DGC between specimens and species. The authors stated above that carbon dioxide can be buffered (see line 88) and thus does not represent the actual carbon dioxide output at steady state conditions. Therefore I doubt that carbon dioxide release rate (which has to be measured over many complete cycles in order to get a significant result) is a good parameter. Furthermore, there is no measure of conductance of the tracheal system and spiracles, tracheal volume and buffer capacity of the tissues and hemolymph given in the datasets. These may vary from "burst" to "burst" and may affect pattern generation and in turn the outcome of carbon dioxide release.

Response: The reviewer has highlighted an important point. A type that we have embraced and upon afterthought required the inclusion of a paragraph to highlight the limitations of this meta-analysis given the data used. Consequently, we have stated the nuances of the data and suggested ways to improve should a subsequent meta-analysis be desired. Kindly see lines 602-609

169

I doubt that the selection of files which only measure carbon dioxide release rate may help in answering these questions of the ultimate reason why insects close their spiracles.

Response: We highlighted in Lines 229-232 how with consultation with another prominent researcher, we rated metrics that could be used to answer this question – one of which included carbon dioxide release rate. In line 233, we then gave a rationale as to why we decided to go with carbon dioxide release rate. To reiterate, it is widely used and reported. What we didn’t say is that it is the only reason why insects close their spiracles. We provided other context

174

The goal of the study is well defined, but in my opinion the title and the objectives are a little bit out of focus.

Response: This comment is noted

Figure 1

Since not everyone knows these type of recording: It would be great to use the same axes-scaling in order to highlight the variation in amplitude and frequency of these events.

Response: These were illustrations excised from published works of literature. We do not have permission to rescale

Figure 2

I like this diagram

Response: We appreciate and thank the reviewer for their kind word

227

I can see that the authors used some prominent features of the DGC which the reviewer also regards as important. There are, however, no criteria on the magnitude of these

Response: This comment is noted

231

I agree with the flow rate being important in order to evaluate the time response of the system. But this also largely depending on volume of the respirometer chamber and tubing length. At least the recording devices seem uniform throughout the studies selected.

Response: This comment is noted

241

Good. I see that combining data into a interburst data set are an appropriate way to overcome the limitations of deciding which phase in the DGC-cycle we are talking about.

Response: We appreciate and thank the reviewer for their kind word

245 

In you data set the body masses spread over a relatively big range (in obj2) from ~30mg to nearly 2g. Unfortunately the body masses for obj3 are not shown but may vary by around 2 orders of magnitude too. Did you take into account the nonlinear relation between body mass and mass specific metabolic rate (µl/g min) in your further evaluation considerations? A small animal (e.g. 20mg) with higher metabolic rates may probably face the same problem as a 2g beetle having a lower mass specific metabolic rate? You could have included this into a log-lin model like the T/MR-Model.

Response: Yes, we did. Not only that. the model accounted for a number of variations including order, family, and species

245 and 254

Wouldn´t it have been more appropriate to convert both data on carbon dioxide emission and water loss rate into molar fluxes (e.g. µmol/g h) instead of converting one of them into a volume flow (µl/g min) and the other one into a mass flow (mg/min) and NOT converting it into a mass specific rate (mg/g min)? In my opinion this procedure could have given more insights like the ratio of water loss to carbon dioxide release rate or a transpiration vs. respiration quotient which might be a suitable measure of water saving. Please reconsider.

Response: The data were reported in the papers we used in volume and mass units, not moles. People understand and can relate to volumes and masses. Yes, we could convert the data, but we would probably be less accurate because we would have to make assumptions for the conversion: the data we used have already been processed. Importantly, even if we followed through with the conversion, we would be converting final values since smoothing, averaging, and integration has already been done. Again, it won’t really make and difference. Our study isn’t really a mechanistic study. It’s an analysis of other studies to try to find support for one or another hypothesis.

263

You mentioned the variability in experimental procedures. This is a critical point as I expressed above. A short section on this problem would be of big use for the introduction part of the work.

Response: True. We have included this limitation in Lines 602-609

325 and fig. 3

I am not really surprised by the finding.

351

I am surprised that the DGC was not shown under hyperoxia. This may come from a limited dataset and the selection of species which do not live in habitats with low O2 and high CO2 mentioned by the original hypothesis. Also the variability and selection of either pupae, larvae and adults and a wide range of weight may affect these results. Also, I wonder whether the maximum carbon dioxide release rate rather than the average (mean?) would be a more appropriate measure to be included. I was not able to check all cited data for this relation.

Response: True. We have included this limitation in Lines 602-609

387

It is hard for me to figure out what significance this result has for the initial question of the paper.

Response: It doesn’t directly. It was an unexpected and interesting result that we thought should be reported.

404

This is one crucial point. The role of single spiracles in the DGC which has not been asses here.

Response: It is almost impossible for one single paper to try to cover every possible base. Even so, we have embraced this point and offered it as parts of points for subsequent work to advance. Kindly see Line 602-609

413

I would not call this a simple advance in technique. Many methods applied in the older work from Schneiderman, Beckel, Levy, Kanwisher were very important. These techniques (pressure, O2/CO2 concentration/abdomen movement) have not been developed like flow through respirometry (which has the advantage of high temporal resolution) but respirometry is calibrated on these older techniques. I would suggest a more appropriate wording.

Response: We beg to differ with the reviewer. As duly pointed by the reviewer, ‘current generation of respirometry was calibrated on older techniques’. This clearly reflects an advancement in technology. These new tools are much more sensitive and measure O2 consumption and CO2 production rates directly and not indirectly like older tools. In other words, these new tools are more technological advanced than the older ones. Importantly, the wording we used is “advances in technology” (see Line 416) not “advances in technique”

419

One could draw the alternative conclusion that these small scale experiments should be extended into other arthropod groups involving the same techniques, methods, protocols and the same environmental parameters.

discussion

The discussion section is great which came a little bit surprising for me after I read through all the previous parts of the paper which made me think of limitations in answering the initial question of the study. It shows a lot of methodological problems (which I mentioned in parts before) and a lack of data which - in my opinion can not only be gained by flow through respirometry and/or broad scale evaluation of data from other taxa.
Especially interesting is the question for any measurable fitness parameters which could be an important evolutionary drive in eco-physiological adaptation. Comparing animals from different collection sites (with crossing experiments) could also advance the knowledge about the significance of the DGC. Testing heritability (morphologically, physiologically) would be a great experiment.

Response: We agree with the reviewer. Thus, we have included a paragraph that discusses these limitations in Lines 602-609

Round 2

Reviewer 1 Report

Thank you for correcting the manuscript. Please, invert the order of references 63 and 64.

Reviewer 2 Report

All necessary improvements to the manucripts have been made. Thank you!